# Comparison of Antifungal Activity of *Bacillus* Strains against *Fusarium graminearum* In Vitro and In Planta

**DOI:** 10.3390/plants11151999

**Published:** 2022-07-31

**Authors:** Catherine Jimenez-Quiros, Emeka C. Okechukwu, Yiguo Hong, Ömür Baysal, Mahmut Tör

**Affiliations:** 1Department of Biology, School of Science and the Environment, University of Worcester, Worcester WR2 6AJ, UK; c.jimenezquiros@worc.ac.uk (C.J.-Q.); e.okechukwu@worc.ac.uk (E.C.O.); y.hong@worc.ac.uk (Y.H.); 2Research Centre for Plant RNA Signaling, College of Life and Environmental Sciences, Hangzhou Normal University, Hangzhou 311121, China; 3Department of Molecular Biology and Genetics, Faculty of Life Sciences, Muğla Sıtkı Koçman University, 48121 Muğla, Turkey; omurbaysal@mu.edu.tr

**Keywords:** *Fusarium graminearum*, *Bacillus*, biological control, mycotoxin

## Abstract

*Fusarium graminearum* (*Fg*) causes Fusarium head blight (FHB) disease in wheat and barley. This pathogen produces mycotoxins including deoxynivalenol (DON), the T-2 and fumorisin B1. Translocation of the mycotoxins in grains causes important losses in yields and contributes to serious health problems in humans and livestock. We tested the *Bacillus* strains, two commercial, QST713 (Serenade^®^) and FZB24 (TAEGRO^®^) and one non-commercial strain EU07 as microbial biological control agents against the *F. graminearum* strain *Fg-*K1-4 both in vitro and in planta. The EU07 strain showed better performance in suppressing the growth of *Fg*-K1-4. Cell-free bacterial cultures displayed significant antagonistic activity on *Fg*-K1-4. Remarkably, heat and proteinase K treatment of bacterial broths did not reduce the antagonistic activity of *Bacillus* cultures. DON assays showed that *Bacillus* strain was not affected by the presence of DON in the media. Leaf and head infection assays using *Brachypodium distachyon* (Bd-21) indicated that EU07 inhibits *Fg-*K1-4 growth in vivo and promotes plant growth. Overall, the EU07 strain performed better, indicating that it could be explored for the molecular investigations and protection of cereal crops against FHB disease.

## 1. Introduction

In the history of humankind, cereal crops have been important as they are staple foods. It is calculated that about 29% of the world production of cereals corresponds to wheat, which was around 757.4 million of tonnes for the period of 2017-18 [1]. However, worldwide population is estimated to reach 9.2 billion by 2050 and if recent trends continue, the number of people that would be affected by hunger could surpass 840 million by 2030. To prevent this, the United Nations established zero hunger as Sustainable Development Goal 2 (SDG2) [2]. In general, the production of cereal crops is typically affected by cultural practices, germ lines selected, droughts, climate change and plant diseases and pests.

*Fusarium graminearum* (*Fg*) is a major destructive pathogen that reduces global yield of cereals. It causes Head Blight (referred to as Fusarium Head Blight-FHB) in wheat and other cereal crops [3]. It infects the grains in the anthesis stage, when the grains are still soft, and the humidity of the environment is adequate. As a result of *Fusarium* infection, the grains shrink and decay leading to subsequent decrease in yields in the fields. In addition, this fungus produces mycotoxins that can be lodged in the grains. These mycotoxins are highly stable and can persist until the grains are processed for human or livestock consumption [4,5,6]. The main mycotoxins in FHB are deoxynivalenol (DON) and T-2 toxin (T-2), found alongside zearalenone (ZEN) and fumonisin B1 (FB1). These mycotoxins are toxic to human and animals and can be deadly if consumed at high quantity [7,8].

Multiple efforts have been made to control *Fg* and its mycotoxin production. The strategies to control it include crop improvement, crop management, application of chemical pesticides, RNA silencing, DON degradation and use of Microbial Biological Control Agents (MBCA) [9,10,11,12,13,14,15,16]. The use of fungicides to control fungal pathogens was the first option for some years ago [17]. However, environmental issues, occurrence of fungal resistance to these pesticides and accumulation of chemical residues on crops make the use of fungicides unfavourable in agriculture. Some of the fungicides including Carbendazim and Epoxiconazole have already been banned in the UK (Health and Safety Executive-HSE) and other countries. On the other hand, more than 44 *R-*genes have been used in wheat breeding [18]. However, it is recognised that the improvement of cereal crops through new and better *R*-genes came with challenges. The most important issue is the breakdown of resistance by the pathogen due to pathogen mutability. Crop rotation is another option to control FHB. [19] described how the contamination by FHB is more severe when the preceding crop is maize, durum wheat or oats, rather than wheat or barley.

More recent approaches in the control of agriculturally important pests and pathogens include the use of small RNAs (sRNA), also known as spray-induced gene silencing (SIGS) strategies. These are small double stranded RNAs that can target pathogen’s mRNA and induce protection against pathogens. [20] used this strategy to protect barley against *F. graminearum* genes ARGONAUTE and DICER protecting barley leaves. The possibilities of the technique are attracting the interest of many stakeholders in the agriculture circle [21].

The use of Microbial Biological Control Agents (MBCA) against *F. graminearum* has been suggested [22]. Bacillus strains represent a vast majority of the efforts to control this fungus [9,13,23,24,25,26].

The bacillus group includes *B. subtilis*, *B. amyloliquefaciens*, *B. licheniformis*, and *B pumilus*. These bacteria have the characteristics of forming a biofilm, which is composed of the secretion of extracellular polymeric substances (EPSs) that allows the bacteria to aggregate and adhere each other or to surfaces. The biofilm serves like a matrix for interchanges of micronutrients and for communication between cells [27]. Interestingly, these *Bacillus* species are difficult to differentiate using only the 16S rRNA gene because all of them belong to the *B. subtilis* species complex. Nonetheless, in recent decades many other species have been included in this complex group and the use of whole genome sequencing (WGS) contributed to the differentiation of species. On the other hand, an operational group called *B. amyloliquefaciens* was also included in the *Bacillus subtilis* complex group, [28].

Several studies suggest that *Bacillus* strains, such as *B. amyloliquefaciens*, *B. mojavensis* and *B. velezensis* produce antimicrobial compounds including fengycin (Fen D), bacillomycin (Bmy A) and Iturin (Itu C); non-peptidic antimicrobial compounds including polyketides, aminosugars and phospholipids; signal peptides of pheromones and bacteriocins polyketides; amylocyclicin, bacillaene and difficidin that are able to inhibit the growth of *Fusarium* species, [9,27,29,30,31]. In addition, compounds such as fengycin and surfactin and other antimicrobial peptides make the plasma membrane of the hyphal cells more permeable, thereby impeding their growth [32].

In addition, *Bacillus* sp. have the capability to produce volatiles, which induce deterioration of *F. oxysporum* hypha [29].

In this research, three *Bacillus* strains: two commercial (QST713, FZB24) and one non-commercial strain (EU07) were used to investigate their ability of *Bacillus* sp. to control the mycotoxin-producing fungus *F. graminearum* at the in vitro and in planta level. To understand this, a set of antagonisms assays were carried out, in an increased complexity fashion, to determine if there are inherent differences in their capacities to control *F. graminearum.*

## 2. Materials and Methods

### 2.1. Microorganisms Used and Culture Media Conditions

*F. graminearum* (*Fg*) isolate K1-4 (*Fg*-K1-4) was obtained from Paul Nicholson (The John Innes Centre, UK) and used throughout this study. Three *Bacillus* strains, QST713, FZB24 and EU07 were used in this research [29,33]. The fungus (*Fg*-K1-4) was maintained on potato dextrose agar (PDA) medium at 22–24 °C. Occasionally, the fungus was subcultured on spezieller-nährstoffarmer agar (SNA) media or 25% of strength of PDA to reactivate the production of macroconidia [3]. The Bacillus strains were maintained on NA medium at 28 °C. For a long-term storage, the fungus was grown on PDA media around 7d and later sterile pieces of Whatman 01 filter paper (4 mm diameter) were placed over the fungus and allowed the fungus to grow over them for around 7d. Eventually, the filter papers were removed and enclosed with small sterile envelops and stored at −80 °C. The *Bacillus* strains were maintained on nutrient agar (NA) medium at 28 °C. However, to produce culture broth, *Bacillus* strains were grown in LB broth (72 h, 28 °C, 200 rpm). In order to produce glycerol stocks, equal volume of Bacillus culture and 50% glycerol was mixed and aliquoted in cryogenic tubes. The tubes were labelled, flash-frozen in liquid nitrogen and stored at −80 °C for further use.

### 2.2. Verification of Bacillus Strains by PCR and Sequencing

All the isolates were subjected to PCR amplification using the universal 16S primers UP 27F/1525R (5′-AGAGTTTGATC(M)TGGC-TCAG-3′ and 5′-AAGGAGGTG(W)TCCA(R)-CC-3′) and colony PCR technique. BioMix™ Red (Meridian Bioscience, London, UK) was used and the PCR regime employed was initial denaturation at 96 °C for 5 min, final denaturation at 94 °C for 30 s, annealing 56 °C for 30 s and initial extension at 72 °C for 1 and final extension at 72 °C for 10 min for 30 cycles. The PCR products were run in a 1% agarose gel to determine if the PCR worked. PCR products were then cleaned up and sent for Sanger sequencing. BLASTN analysis of sequence results obtained against the known Bacillus 16S rRNA gene sequences in NCBI databases were performed [34]. The isolates corresponding to *Bacillus* subtilis were kept and used in the experiments.

### 2.3. Antagonism Assays Using Bacillus Plugs against Fg-K1-4

With the aim to demonstrate whether the bacteria have the ability to suppress the fungal growth, antagonism assays were carried out using the *Bacillus* strains (QST713, FZB24 and EU07). Petri dishes with PDA/NA were used for in vitro antagonism assays. Sterile plug cutters (10–15 mm) were used to obtain circular pieces of fungus and *Bacillus* strains. After placing the fungal and bacterial plugs on the media in the same Petri dishes, plates were wrapped with cling film to avoid external contamination. All the assays were performed at 28 °C incubator for 4d (Sanyo Biomedical Co., London, UK). After 4d, pictures were taken using a camera in its stand. All the pictures were taken with the camera (Canon EOS-350D) in its stand at the same height, focus and resolution to provide the uniformity. The images were processed and analysed with the ImageJ software [35]. The scale of the image was set using the diameter of the Petri dish (90 mm) and applied globally. Subsequently, the area of the fungal growth was selected and measured. The software brought the output with the areas and then the data were imported into a spreadsheet. The data obtained were processed to obtain the percentage of inhibition for every data point of the experiments [36] and used for the subsequent data analysis. Each experiment had 4 biological replicates and each experiment was repeated at least three times.

### 2.4. Bradford Assay

To determine the total protein concentrations in *Bacillus* broths, a Bradford assay was performed using Bradford Reagent (B6916, Merck, Dorset, UK). The experiment was performed in a 96-well plate based on instructions of the manufacturer. Briefly, 1 µL of broth was added to the well containing 250 µL of Bradford reagent for each sample. Each sample had 3 replicates. Additionally, known concentrations of Bovine Serum Albumin (BSA) (1.0, 0.8, 0.6, 0.4, 0.2 and 0.0 mg/mL) were used. The reaction was incubated for a minimum of 10 min and the samples were recorded using a plate reader (Multiskan Go, Thermoscientific, Loughborough, UK) at 595 nm. Absorbance data for the BSA samples were obtained and transformed (net absorbance) and a plot was constructed with the data, as well the tendency line and the equation (y = mx + b). This was used as a standard curve to determine the concentration of total proteins in the broths.

### 2.5. Antagonism Assays Using Whole Bacillus Broths against Fg-K1-4

Antagonism assays of the whole bacterial broths at several dilutions (10^2^, 10^3^, 10^4^, 10^5^, 10^6^ and 10^7^ in water) against *Fg*-K1-4 were carried out to evaluate the interaction of the two microorganisms on PDA/NA media. Serial dilutions were subcultured in NA medium overnight, and the cultures were used for the experiments. Two plugs were cut out from the PDA/NA plates at equal distance from edge of the Petri dish in an opposite position to each other. One plug of *Fg*-K1-4 was placed on one side while the plug of *Bacillus* strains was on the other side of the same plate using sterile tweezers. The assay was evaluated using ImageJ software as described above. Each experiment had 4 biological replicates and was repeated at least three times.

### 2.6. Antagonism Assays Using Filtered Bacterial Broths against Fg-K1-4

After determining that the *Bacillus*–*Fg-*K1-4 interaction was very effective, we decided to use only the bacterial broths in subsequent antagonism experiments. To assess the antagonism effect of broths from the *Bacillus* strains against *Fg*-K1-4, the bacteria were cultured in liquid broth at different time intervals (6 h, 24 h, 48 h and 72 h). The bacterial samples were centrifuged, and the aqueous phase was filtered using 0.22 µm filters (EMD Millipore Millex™, Merck, Dorset, UK). One hundred µL solution was spread over the medium (PDA/NA) using an L spreader and allowed to be absorbed by the medium for around 30 min after which a plug of the fungus was placed on the middle of the plates. The assay was evaluated using the ImageJ software. Each experiment had 4 biological replicates and was repeated at least three times.

After determining that the bacterial broths showed an antagonistic effect on *Fg*-K1-4, an assay using adjusted proteins (total) was performed to evaluate the difference on the antagonistic effect between the *Bacillus* strains. Bacterial broths were cultured for 24 h, 48 h and 72 h, spun, filtered (0.22 um) and assessed with a Bradford assay to determine the total protein concentration. Subsequently, the broths were adjusted and 200 µL at 100 µg/mL of protein was added to the plates, spread and allowed to be absorbed over the time of 30 min. The assay was run over 4d and evaluated according to the above sections. Each experiment had 4 biological replicates and was repeated at least three times.

### 2.7. Heat and Proteinase K Treatment of Bacterial Broths

Due to the antagonistic effect of the adjusted-filtered *Bacillus* broth on the *Fg-*K1-4, a further assay was performed to determine whether the antagonistic effect was derived from a protein component. To investigate this, *Bacillus* broths were centrifuged, filtered and autoclaved (121 °C, 1 atm, 15 min). A Bradford assay was performed to normalise the protein content before and after the autoclaving process and the pre-autoclaved concentration was used to adjust the broths at 100 µg/mL total protein. The effects of autoclaved and non-autoclaved broths on *Fg-*K1-4 were assessed over 4d. A set of new experiments was performed to determine the best concentrations of total proteins that could inhibit the fungal growth. Bacterial broths were adjusted to 0.5, 1.0 and 2.5, 10.0 and 20.0 µg/mL against *Fg*-K1-4 and antagonism assays were carried out. The same steps were also carried out with autoclaved samples. Bacterial broths were subjected to proteinase K treatment to assess the stability of the solution and their protein components. Bacterial broths were centrifuged, and Bradford assays were carried out, samples were treated with 100 µg/mL of proteinase K for 2 h at 37 °C and used in further assays. Subsequently, these proteinase K-treated broths were added to the PDA/NA medium at 1.0 µg/mL and autoclaved. An antagonism assay was performed. The effect was evaluated by using the ImageJ software.

### 2.8. Deoxynivalenol (DON) Assay

DON assay was performed to assess the viability of the *Bacillus* strains on different concentrations of the mycotoxin on plate, which could be correlated to the exposition of the bacterium to the fungus in the fields [37,38]. DON (Sigma D0156) was dissolved in acetonitrile to produce a stock solution (1 g/mL).

Firstly, the minimal amount of inoculum was established in a 96 wells plate, 200 µL of LB broth was added per well, then 1 and 5 µL of bacterial broths (24 h) were added and incubated at 28 °C with agitation (VXR basic Vibrax^®^) for 24 h. Then, the plates were inspected, and data were obtained using a plate reader adjusted to 600 nm wavelength (Multiskan Go, Thermoscientific) to determine the relative concentration of the bacteria in the different treatments. A concentration of 70 ppm of DON is considered toxic for microorganisms. From the stock solution, different amounts of DON solution were added to reach a final concentration of 0, 5, 15, 25, 50 and 70 ppm. The experiment was set up with three replicates per concentration for each *Bacillus* strain, LB broth with acetonitrile alone and untreated *Bacillus* in LB broth (controls). The plates were left on the safety cabinet for around 30 min to allow the evaporation of the acetonitrile, then incubated at 28 °C with agitation (VXR basic Vibrax^®^) for 24 h.

### 2.9. Root and Head Infection Assays with Bd-21

*Brachypodium distachyon* (Bd-21) seeds were obtained from Paul Nicholson (The John Innes Centre, UK) and used throughout this study. For root infection assays, sterilised water agar (WA) media were prepared (1.5% agar in distilled water) and poured on square plates (100 mm, non-compartmentalized) allowing the plate to be filled up to 50% capacity (~70 mL of medium). Then, a row of 6–7 seeds were placed over one side of the plate. The seeds were placed parallel to the bottom and sank at least 50% on the medium with an artist brush oriented towards the top. WA plates with seeds were allowed to develop roots until they covered the third part of the plate. Then, a drop of *Bacillus* strains or *Fusarium* suspension (10 µL) was placed through the medium near the root, taking care not to cause damage on the root stem. Pictures of the plates were taken every day for the period of 10 days to observe the growth of either microorganism.

For head infection assays, Bd-21 seeds were vernalised for 2–3 weeks, then sown on pots. Plants were allowed to grow until spikelets (8–10 weeks) were developed and flowered. For inoculations, filter paper (Whatman 01) pieces (2 × 4 mm) were placed carefully between two florets per spikelet, with a total of five spikelets per plant. Then, a drop (10 μL) of fungal spore suspension (10^4^/mL) was applied to the filter paper and allowed to dry ~30 min [39]. Subsequently, some plants were treated with bacterial suspension (EU07, 10%) and allowed to dry in the same manner. Plants were then placed back in growth room (21 °C, 10 h D/14 h L) and covered individually for 3 days with polythene plain grip seal bags to allow a high humid environment, which promotes the fungal growth. After this period, covers were removed, and plants were assessed in the presence or absence of disease symptoms.

To minimise the impact of experimental error in the final statistical analyses, a scale of severity of the infection were constructed with a wide range of disease scoring −6 points [39,40]. The scale ranges from 0 to 5 in increasing order of disease severity. The extent of severity of the disease was used for the scoring. Assessments were carried out every week for 2–4 weeks.

### 2.10. Detached Leaf Infection Assay with Fg-K1-4

Leaves (<9 cm long) were cut at the collar level and sunk in distilled water to keep them turgent. Glass rods with round tip were sterilised individually and used to gently scratch the cuticle of the leaf blade at 5 cm of distance from the collar to produce a wound on the tissue [39]. A 10 μL drop of water-control or fungal solution (10^6^ spores/mL) were deposited over the wound and allowed to dry for about 10–20 min. Treated leaves were taken by the collar and inserted into the WA medium, 6–7 leaves on a row, covered with the lid, labelled and sealed. Plates were stored in a growth room (21 °C, 10 h D/14 h L) for up to 8 days to allow the leaves to develop infection. In a similar way, leaves were taken and sunk in water or bacterial suspension (10%), placed on a towel and allowed to dry for ~10 min. Later, mechanical damages were made on every leaf as described above and leaves were treated with spore solution, placed on WA medium and incubated on growth room for a period of up to 8 days. Pictures were taken of the development of the infection. At the end of the experiment, the infected section was cut (above the infected area) and treated with trypan blue.

### 2.11. Trypan Blue Staining

To reveal hyphal structures and dead plant cells in plant tissues, trypan blue staining was used. Sections of infected leaves of Bd-21 were cut at 2 inches long and stained with trypan blue. Phenol, glycerol, lactic acid, water (1:1:1:1) and 0.05% of trypan blue were mixed together as a stock solution and filtered. When needed, the stock solution was mixed with ethanol (>95%) 1:1. Tissues were immersed in trypan solution in a suitable container with loose lid, placed in boiling water for one minute, put aside and rested for 24 h in a fume hood. Tissues were cleared of the trypan solution and de-stained with 40% chloral hydrate. When cleared, tissues were preserved in 50% glycerol.

## 3. Statistical Analysis

One-way or Two-way ANOVAs were conducted using SPSS 25.0 to examine the effects of different treatments. The means (*M*) and standard deviation (*SD*) as well as the pairwise comparisons were reported. Letters were used to show the difference between the means. If two variables have different letters, they are significantly different.

## 4. Results

### 4.1. Bacillus Strains Inhibits Fg-K1-4 Growth

Bacillus strains QST713, FZB24 and EU07 were verified by sequencing and a dual culture assay to determine their antagonistic effect on the *Fg-*K1-4 was set up. Initially, a dilution series of bacterial cultures were tested against *Fg-*K1-4. All strains showed high inhibition of the fungal growth compared to controls (Figure 1A). The percentage *Fg*-K1-4 growth inhibitions by the different *Bacillus* strains were in the range of 73.0–57.9%, 76.7–61.5% and 79.3–62.5% for QST713, FZB24 and EU07, respectively. The highest percentages of inhibition were observed with 10^3^, 10^4^, 10^5^ dilution series with values of 79.3–57.9%, 76.7–62.3% and 75.5–61.6%, respectively. The main effects of dilutions were statistically significant (*p* ≤ 0.018) where dilutions 10^2^ and 10^4^ showed significantly higher suppression than those obtained with 10^7^ dilutions. Additionally, the strains EU07 and FZB24 showed significantly higher % inhibition (*p* ≤ 0.001) than QST713 (Figure 1B).

### 4.2. Cell-free Bacterial Supernatants Have a Suppressive Effect on Fg-K1-4

We wanted to determine whether the inhibition of fungal growth observed above was due to the presence of bacteria or the bacterial products; we therefore carried out antagonism assays using cell-free cultures. Initially, cell-free bacterial supernatants (CFS) were obtained at 6 h, 24 h, 48 h and 72 h after culture initiation from the *Bacillus* strains and filtered. Subsequently, the filtered CFS were used to assess the inhibition effect on the *Fg-*K1-4. Results reveal that the treatment with CFS from all the bacterial cultures that was tested at 6 h did not have any significant antagonistic effect on the fungus (Appendix A).

However, in comparison to 6 h (−2.7–0.5%), filtrates of the bacteria at 24 h and 72 h had the higher levels of inhibition with a range of 36.6–39.0% and 31.6–46.7%, respectively. On average, the broth from the strain FZB24 showed the highest % of inhibition (30.5%) (Appendix A). An analysis of the main effect for culture times showed a statistically significant (*p* ≤ 0.011) difference. Similarly, the analysis of the main effect for the *Bacillus* strains showed statistically significant difference (*p* ≤ 0.037). We then normalised the protein in bacterial cell free broths to 100 µg/mL using Bradford assay and repeated the experiments using culture broths obtained at 24 h, 48 h and 72 h (Figure 2A). Our results show that growth inhibitions of *Fg-*K1-4 by culture time were 5.73–65.38% (24 h), 58.33–70.07% (48 h) and 69.28–75.60% (72 h) by the strains QST713, FZB24 and EU07, respectively.

A two-way ANOVA (after square root transformation) showed that there was a statistically significant interaction between *Bacillus* strains and culture times on % growth inhibition of the *Fg*-K1-4, (*p* ≤ 0.001). There was a statistically significant difference in % inhibition of the *Fg*-K1-4 by the *Bacillus* strains, with *p*-values of EU07 (*p* ≤ 0.005), FZB24 (*p* ≤ 0.005) and for QST713 (*p* ≤ 0.005). Cultures of the EU07 strain at 24 h had a statistically significantly lower % inhibition of the *Fg*-K1-4 than EU07 strain at 48 h, 1.491 (*p* ≤ 0.001) and 72 h, 2.300 (*p* ≤ 0.001) (Figure 2B).

### 4.3. Heat Treatment of Bacterial Broth Does Not Affect Antagonism on Fg-K1-4

After determining that the filtered-adjusted broths showed significant consistent fungal growth inhibition, the bacterial broths were subjected to heat treatment to understand whether the broths would be able to keep the inhibitory effect. Broths were obtained from 24 h bacterial cultures and adjusted to 100 µg/mL of total proteins, filtered and autoclaved. Antagonism assays were then performed (Figure 3A).

All the autoclaved *Bacillus* broths showed higher antagonistic effect in comparison to non-autoclaved broths. The unautoclaved bacterial broths showed 53.9–73.4% inhibition of *Fg*-K1-4 whereas the autoclaved bacterial broths showed an inhibition within the interval of 74.5–76.0%.

A two-way ANOVA showed that there was a statistically significant difference between *Bacillus* strains and treatments on % inhibition of the *Fg*-K1-4, (*p* ≤ 0.001).

There was also a statistically significant difference in % inhibition of the *Fg*-K1-4 by the *Bacillus* strains at both filtered, adjusted (FA) or filtered, adjusted and autoclaved (FAA), for EU07 and QST713 (*p* ≤ 0.005). Additionally, there was a significant difference in % inhibition of *Fg*-K1-4 for FA for all strains used, EU07, FZB24 or QST713, (*p* ≤ 0.0005).

EU07 strain for FAA had significantly higher % inhibition of *Fg*-K1-4 than EU07 for FA, 21.935%, (*p* ≤ 0.001). FAA from QST713 showed a significantly higher mean than that of FA from QST713, 12.785% (*p* ≤ 0.001). There was no significant difference between the treatments of FA and FAA for FZB24 (Figure 3B).

After determining that the heat-treated broths from *Bacillus* strains had an antagonistic effect on *Fg*-K1-4, further antagonism assays were performed to evaluate different concentrations of proteins in the bacterial broths (Figure 4A).

Interestingly, the treatment with 1.0 mg/mL had a similar antagonistic effect (67.58–81.45%) to 2.5 mg/mL (67.79–75.41%) for all the *Bacillus* strains. However, QST713 strain showed the lowest antagonistic effect at 0.5 mg/mL with 17.18% followed by EU07 with 60.59%. Notably, a high variability of the data was observed with the treatment using 2.5 mg/mL broths.

The two-way ANOVA showed that there was a significant difference among the *Bacillus* strains in relation to the % inhibition of *Fg*-K1-4 (*p* ≤ 0.0001). There was a statistically significant difference in % inhibition of the *Fg*-K1-4 by *Bacillus* strains for EU07 and QST713 (*p* ≤ 0.0005) and for FZB24 (*p* ≤ 0.021) at either 0.5, 1.0 and 2.5 mg/mL.

There was a significant difference in % inhibition of the *Fg*-K1-4 by dosage 0.5 mg/mL at whichever *Bacillus* strain (*p* < 0.0005) was used and for treatments of 1.0 mg/mL for any *Bacillus* strains (*p* ≤ 0.0005). No significant differences were obtained in % inhibition of the *Fg*-K1-4 among the *Bacillus* strains using their broths adjusted to 2.5 mg/mL of total proteins (*p* ≤ 0.005) (Figure 4B).

Similar to above experiments, another antagonism assay was performed to evaluate if high concentrations of proteins would completely prevent the fungal growth (Figure 5A,B). The two-way ANOVA showed that there was a significant difference between *Bacillus* strains and dosages (10 and 20 mg/mL) on % inhibition of the *Fg*-K1-4 (*p* ≤ 0.024). Significant difference was obtained in % inhibition by *Bacillus* strain EU07 either at 10 or 20 mg/mL (*p* ≤ 0.009). Similarly, there was a significant difference in % inhibition of *Fg*-K1-4 using 20 mg/mL of EU07, FZB24 or QST713 (*p* ≤ 0.005). No other significant difference in % inhibition of the *Fg*-K1-4 was found on the treatments.

The Tukey-HSD showed that EU07 strain with 10 mg/mL had significantly higher % inhibition than that obtained from EU07 strain with 20 mg/mL, 2.457% (*p* ≤ 0.001). FZB24 strain and QST713 strain did not show any significant differences between the treatments of 10 mg/mL and 20 mg/mL.

Treatment with 20 mg/mL of FZB24 had a significantly higher % inhibition than that observed from 20 mg/mL of EU07, 3.03% (*p* ≤ 0.005). However, the *Bacillus* strains did not differ significantly from one another in suppressing the fungal growth when 10 mg/mL was used.

### 4.4. Proteinase K Treatment of CFS Does Not Alter Antagonism against Fg-K1-4

After determining that even the low concentrations of proteins have a sufficient antagonistic effect on the fungal growth, we tested whether the bacterial broths kept their stability in the presence of proteinase K, which is known to degrade proteins within the given environment (Figure 6A,B). The assay showed that the average of the % inhibition by 2.5 mg/mL and 5.0 mg/mL the treatments was 67.89% and 71.94%, respectively. The QST713 strain showed more variability on the % inhibition for both treatments as compared to other strains.

The interaction effect between *Bacillus* strains and dosages (2.5 mg/mL and 5.0 mg/mL of proteins) on % inhibition was not statistically significant, (*p* = 0.127). Thus, analysis of the main effect (*Bacillus* strains only) was performed, which indicated that there was a significant difference (*p* ≤ 0.003). The treatment with the *Bacillus* strain EU07 was significantly different (*p* ≤ 0.020) from that obtained with QST713 (8.357 higher % inhibition). The FZB24 treatment was higher than QST713 by 10.573 % inhibition (*p* ≤ 0.003) (Figure 6B).

### 4.5. DON Does Not Have Any Effect on Bacillus Growth

As mycotoxin DON is highly soluble in acetonitrile, an initial experiment was carried out to determine the effect of the acetonitrile (Ac) on the growth of the *Bacillus* strains. The results show that *Bacillus* strains could grow well in 96-well plates after inoculation, either with 1 or 5 µL of bacterial broth (24 h old) with a visible biofilm. Likewise, the bacteria can grow on LB medium where Ac was added in the concentrations tested in this study. However, there was a significant difference in the growth of the FZB24 strain in comparison to the other two (F (2, 68) = 7.064, *p* = 0.002) according to the ANOVA test.

DON assays were performed to determine the viability of the bacteria on different concentrations of mycotoxin (0, 5, 15, 25, 50 and 70 ppm).

While the data did not meet the normality test (Shapiro *p* > 0.05), the homogeneity test (Levene’s tes of equality *p* = 0.002) presented outliners (one outliner on EU07 and one in QST713 data). Nonetheless, ANOVA (two-way) is robust enough to tolerate this. Therefore, the analysis was performed.

There was no significant difference between *Bacillus* strains and DON (dosage) with respect to the bacterial growth (as absorbance), F (10, 54) = 1.893, *p* = 0.066, partial η2 = 0.260. The main effects showed that there were significant differences in the growth of *Bacillus* strains, F (2, 54) = 15.565, *p* < 0.0005, partial η2 = 0.366. Besides, there was no significant difference between bacterial growth within the concentrations of DON used, F (5, 54) = 1.115, *p* = 0.363, partial η2 = 0.094 (Figure 7).

### 4.6. Bacillus Suppresses Fg Growth in Planta

As the in vitro antagonism assays with *Bacillus* strains against *Fg-*K1-4 showed a suppression of fungal growth, and EU07 was the best out of the three *Bacillus* strains, we then used EU07 in in vivo assays using *Brachypodium distachyon* (accession Bd-21). Firstly, root infections assays were carried out to test whether EU07 could be used to inhibit the growth of *Fg-*K1-4. The control treatment mostly showed one main root at 11 days after sowing. In contrast, the group inoculated with *Fg-*K1-4 showed growth of secondary roots from the crown. Remarkably, new roots managed to grow to the same length as the first roots at the 4 dpi (Appendix A). When the roots of Bd-21 were treated with *Bacillus* EU07, growth of the bacterium was not observed.

We then carried out detached leaf infection assays, which were shown to be promising to understand the behaviour of the fungus treated with or without bacterial broth of EU07 (Figure 8). Not surprisingly, *Fg-*K1-4 inoculations showed necrosis on the infection sites and visible mycelial growth around the necrotic area in comparison to the controls. The treatment combination (*Fg-*K1-4/EU07), in which the fungus was inoculated first and later treated with EU07, showed necrotic areas. However, in contrast to the *Fg-*K1-4 treatment, the application of EU07 seemed to inhibit mycelial development as there was no visible mycelial growth around the inoculation site (Figure 8A).

To investigate this further, leaves were stained with trypan blue and observed under a light microscope. Control leaves did not show any infection signs, but in some cases the wounded areas showed some necrosis. The *Fg-*K1-4-inoculated leaves showed abundant mycelia on the surface and masses of macroconidia were also observed (Figure 8B). In the *Fg*-K1-4/EU07 combination, less mycelial development and no macroconidia were observed.

We also carried out further studies on *Fg*-K1-4/EU07 combination using Bd-21 spikelets. A scale of severity was constructed to facilitate the evaluation of the infection on plants during the experiment (Figure 9). The scale was made with six assessment points clearly delimited with the aim of lowering the impact of errors due to the evaluator [39,40,41].

Head Blight infection assays were then performed to assess the effectiveness of EU07 to suppress the *Fg* infection. The Disease Severity Index (DSI) was calculated from the study (Figure 10). The experiment was statistically analysed using two-way mixed ANOVA. There was no homogeneity of variances, as assessed by Levene’s test of homogeneity of variance (*p* > 0.05), but there was homogeneity of covariances, as assessed by Box’s test of equality of covariance matrices (*p* = 0.002).

There was no significant interaction between the treatments and time on the % DSI score, F (6, 59) = 1.642, *p* = 0.152, partial η2 = 0.143. The main effect of time exhibited a significant difference in % DSI at different time points, F (1, 59) = 6.182, *p* = 0.016, partial η2 = 0.095. Similarly, the main effect of the treatments showed a significant difference in % DSI at different time points, F (6, 59) = 23.29, *p* < 0.001, partial η2 = 0.703 (Figure 10). It is worth mentioning that during the course of the assessments, it was noticed that infection by *Fg-*K1-4 alone produced more presence of hypha (visible hypha and necrosis) than drenching *Fg-*K1-4-inoculated plants with EU07 (non-visible and necrosis).

### 4.7. Bacillus Strain EU07 Promotes Plant Growth

Assays with the Bd-21 treated with the *Bacillus* strain EU07 showed significant differences on the production of number of heads compared to the non-EU07 treated Bd-21. Plants drenched with the bacterial solution developed much more heads than those not drenched, while plants infected with *Fg-*K1-4 developed a smaller number of heads per plant in comparison to the control plants (Figure 11A,B). They were assessed with one-way ANOVA to determine if the number of heads in Bd-21 plants was statistically different between the groups or treatment combinations. The treatments were classified into seven groups: Control (n = 11), *Fg* (n = 12), *Fg* + *Bs* drop (n = 10), *Fg* + *Bs* spray (n = 10), D + Control (n = 8), D + *Fg* (n = 8) and D + *Fg* + spray (n = 7). There were outliers in the data (D_*Fg*_Spray [1], Fg [2], *Fg*_D_Spray [1]), as assessed using a boxplot. They were kept as they were and considered normal values. Number of heads for *Fg* + *Bs* drop treatment was not normally distributed, as assessed by Shapiro-Wilk’s test (*p* > 0.05). There was homogeneity of variances, as assessed by Levene’s test for equality of variances (*p* = 0.772). The number of heads produced by Bd-21 in the different treatments differed significantly from each other, F (6, 65) = 4.36, *p* = 0.001.

Treating Bd-21 with *Fg* produced 17.00 ± 3.51 heads, a smaller number of heads compared with that of the control only (18.73 ± 3.97) and control + drenched treatments (22.13 ± 4.61). In general, drenched treatments showed higher production of heads: Control+Drenched produced 22.13 ± 4.61 heads, *Fg* + Drenched produced 25.22± 4.5 heads and *Fg* + Drenched + treated with spray of *Bs* yielded 26.67 ± 5.28 heads. Non-drenched plants infected with *Fg* and treated with *Bacillus* in spray (23.6 ± 3.98) or by drops (23.1 ± 5.9) produced more heads than those treated with *Fg* alone but less than all the drenched treatments.

Tukey post hoc analysis revealed that the drenched with *Bs* and infected with *Fg-*K1-4 treatment was statistically significant in comparison to control (*p* = 0.019), to *Fg* (*p* = 0.002), but not to infected treatment and treated with *Bs* (Spray (*p* = 0.850) or drop (*p* = 0.738)) and the drenched treatments (Control (*p* = 0.529) and *Fg* infected (*p* = 0.997)). Conversely, the treatment Fg-K1-4 was significantly different only to the control treatments - drenched (*p* = 0.195) (Figure 11B).

## 5. Discussion

For sustainable agriculture, integrated management strategies for disease control have been steadily increasing and the use of microbial biological control agents (MBCA) is part of this strategy. Fungal and bacterial microorganisms including *Trichoderma*, *Bacillus* and *Pseudomonas* species have been extensively tested against fungal and oomycete plant pathogens as they produce antimicrobial and volatile compounds, and lytic enzymes [42]. In addition, it is well known that the use of plant growth-promoting bacteria (PGPB) provides advantages for the plant growth and development and induction of abiotic and biotic stress tolerance [43].

In these regards, we carried out antagonism assays to evaluate the effectiveness of three *Bacillus* strains against proliferation of *F. graminearum* strain, *Fg*-K1-4. Firstly, the in vitro assay using the bacterial broths showed that all three *Bacillus* strains possess antagonistic effect against *Fg*-K1-4, but EU07 strain was more consistent in the suppression. Since the dual culture assays indicated clear antagonistic effects, we further investigated the effect of different concentrations of the bacteria on the inhibition of fungal growth. The result in general shows that the *Bacillus* strains exhibited a high percentage inhibition of ~80.0% at the concentration of 10^3^ and 10^4^, which agrees with those reported in other similar studies [23,44].

Furthermore, we tested the cell-free bacterial supernatants (CFS) from the *Bacillus* strains against *Fg*-K1-4 as the bacteria can be temperature-dependent, and CFS can contain some metabolites that could act as bio-stimulants and biocontrol agents [29,43,45]. This assay in general might not display total effectiveness of the *Bacillus* strains compared to using the whole bacterial culture. Nevertheless, CFS showed antagonistic effects on *Fg*-K1-4. Data collected using CFS from 72 h culture exhibited higher antagonistic effect, probably due to high accumulation of antagonistic compounds in the media, but the samples at 24 h showed consistent and similar antagonistic effect among the *Bacillus* species. This could be that the 72 h-culture had a higher number of bacterial cells formed, leading to higher build-up of antifungal compounds. [46] found that the growth curve of the *B. subtilis* reached the exponential phase after 18–24 h or 30 h (according to the strain used) in LB broth and afterwards it reached the stationary phase. It has also reported that some antifungal compounds such as iturin A, lipeptides and fengycins could be detected in culture grown up to 72 h using High Performance Liquid Chromatography [32,46].

Thus, the above-mentioned assay prompted us to use CFS with adjusted concentration of total proteins. Surprisingly, adjusting the protein concentration of the CFS for each strain exhibited a higher percentage inhibition of the fungus than the previous assay (non-protein-adjusted CFS) and similar to the dual culture assay (using the *Bacillus* broth against the fungus). In this case, the strain EU07 as well the FZB24 showed the best results. The strain FZB24 was superior to EU07 in their antagonistic effect against the *Fg*-K1-4 when the concentration of proteins was adjusted and sampled at 24 h.

It is clear that the antagonistic effect observed is due to molecules secreted by the *Bacillus* strains in the medium, and not only due to the rapid growth of the bacterial cells. Notably, the secretome of the *B. subtilis* strain 168 has been studied extensively for its ability to produce many compounds that are beneficial for industry [29,47,48]. In addition, treatment of CFS with heat and proteinase K did not decrease the activity of the CFS against *Fg*-K1-4 indicating that the antagonistic compounds are thermostable and not proteinaceous but possibly antimicrobial compounds such as fengycin and iturin A [49]. In other studies where compounds were purified, the minimal inhibitory concentrations were found to be in the range of 50 to 128 µg/mL for iturin A and fengycin A [24,45,50]. This perhaps suggests that it is necessary to use higher volumes of whole bacterial broths to completely inhibit the mycelial growth of *Fg*.

Interestingly, others have used heat treatment and proteinase treatments on *Bacillus* extracts—Bacteriocin, Protein LCI gene, LCI protein, respectively, to assess their antagonistic effects [51,52,53]. However, their efforts were negative or null when proteinases and autoclaving conditions were used. However, an improved antagonistic effect was obtained after treating CFS with similar conditions in our experiments. It is possible that the conserved antagonistic effect is due to compounds that are degraded exposing active sites of antimicrobial molecules, which led to the release of a more potent compound in the media after the treatments.

It has been reported that Deoxynivalenol (DON) has a crucial protagonist role in the development of the FHB disease [4]. Few studies explore the toxicity of the DON on microbiota and the efforts are more directed towards the detoxification of the mycotoxin using bacteria. The approaches are highly variable, ranging from detoxification of the feed to detoxification on the guts of the animals, to detoxification of infested soils. Many efforts have been carried out to understand how to detoxify food contaminated with high concentrations of DON [10,54,55]. In this study, we were interested in determining if the three *Bacillus* strains could survive on high levels of toxicity of DON. *Bacillus* strains were exposed to different concentrations of DON (0 to 70 ppm) and it was found that the *Bacillus* strains displayed normal growth in presence of the toxin with production of biofilms after 24 h. These results are remarkable due to the consideration that [56] reported that the *Fg* strain 5035 produced 57 μg/g of DON and by legislation in the EU, the maximum level of DON accepted in feed products is 2 ppm (2 µg/g) [8].

Having shown that the *Bacillus* strains can be used to control *Fg* in vitro, further antagonism assays were performed in planta using *Brachypodium* plants (Bd-21). The plant has been reported to correlate well with the cereal crops in response to *F. graminearum* [14,39]. Root infection assay revealed that *Fg-*K1-4 did not distort the shape of the roots, but the plant responded with production of secondary roots from the crown point. Some suggest that the *Bacillus* QST713 has genes that enable it to colonize plant roots [57], however, we did not observe this capability for any of the *Bacillus* strains on Bd-21 roots.

Furthermore, the effect of the *Bacillus* EU07 was investigated in the leaves infected with *Fg*-K1-4. Results confirm that the leaves infected with *Fg-*K1-4 produce lesions with necrotic areas [39] and occurrence of mycelia was also observed. Interestingly, leaves infected with *Fg*-K1-4 and treated with EU07 displayed little or almost no visible mycelia (Figure 8). Conversely, the *Fg-*K1-4-infected leaves (without EU07) showed a high prevalence of mycelia and spots with macroconidia. *Fg-*K1-4 can develop macroconidia as early as 48 h [58]. The absence or little appearance of visible mycelia observed in the fungal-infected leaves treated with EU07 was in line with [24] who reported that strain S76-3 of *Bacillus amyloliquefaciens* inhibits the *Fusarium* conidia germination completely. Furthermore, [24] explained that the positive antagonism was due to the presence of the lipopeptides iturin A (50 μg/mL) and plipastatin A (100 μg/mL) that causes deformation of the macroconidia and hypha. Thus, the absence of macroconidia formation by *Fg-*K1-4 in the leaf infection assays could be due to the similar activities exerted by the EU07.

Similarly, infection assays in floral structures (heads) were carried out with EU07. The disease severity index (DSI) was calculated by using a scale of severity. Our results show high % DSI in drenched plants, however the progression of the disease was lower than that observed with the treatments without drenching and much lower than the plants infected with *Fg* without any bacterial treatment. During head infection assays, we noted that EU07 has PGP activity. The experiments showed a marked difference in the number of heads produced by plants drenched with the EU07 strain (Figure 11) in contrast with plants sprayed with the *Bacillus* or only infected with *Fg-*K1-4. To confirm this, further assays were carried out including the controls in drenching assay. There seems to be a significant increase in the number of heads infected with *Fg-*K1-4, drenched and sprayed with EU07 (D + Fg + Spray) compared with the control (no drenched) and the infected plants (*Fg*-K1-4 only).

Previous studies demonstrated that EU07 increases plant height [33]. LC-MS analysis showed that EU07 produces 3-hydroxy-2-butanone (acetoin) by VOCs (acetoin or 3-hydroxy-2-butanone), which could help triggering the plant growth [29]. The production of acetoin and 2,3-butanediol by plant growth-promoting bacteria (PGPR) suggests having the induction of systemic disease resistance and drought tolerance [59]. 1-aminocyclopropane-1-carboxylate (ACC) synthase converts S-adenosylmethionine (AdoMet) into ACC, which converts to ethylene by ACC oxidase. This catalytic pathway can be associated with the PGPR activity, which enhances the plant growth by lowering a plant’s ethylene levels as precursor of ethylene; ACC is hydrolyzed by bacteria-expressing ACC-deaminase activity [60].

Overall, our studies here indicate that non-commercial EU07 is superior to other two commercial strains in suppressing *F. graminearum* and is a promising biocontrol agent to use against *Fg*-associated cereal diseases in an effective and environmentally friendly manner in agriculture.

## Figures and Tables

**Figure 1 plants-11-01999-f001:**
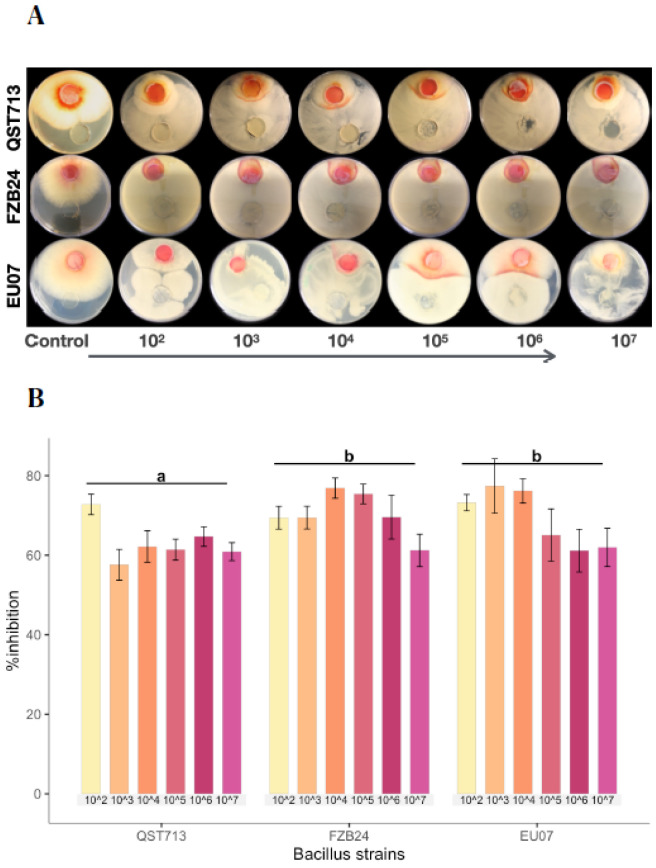
Dilution of *Bacillus* strains inhibits *Fg*-K1-4′s growth. (**A**) Dual culture experiment was set up on PDA/NA with *Fg-*K1-4 (reddish plugs) and the picture was taken 4d after incubation at 28 °C. (**B**) Percentage growth inhibition of *Fg*-K1-4 by the *Bacillus* strains with different dilutions. Data were from one independent experiment with four repeats shown and the mean ± SE were displayed. Experiment was replicated at least three times with similar results. Bars clusters with different letters were significantly different according to Tukey’s Test (α < 0.05) following two-way ANOVA. n = 72.

**Figure 2 plants-11-01999-f002:**
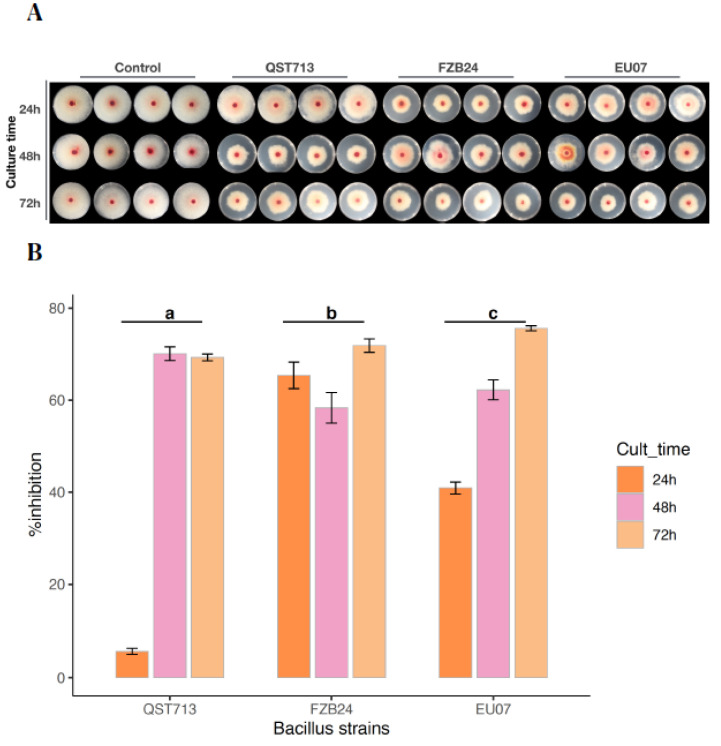
Normalised bacterial broths of *Bacillus* strains suppress fungal growth. (**A**). *Fg*-K1-4 growing on PDA/NA medium containing filtered and adjusted bacterial broths. The fungus was grown on PDA/NA medium in the presence of the 100 µL filtered and adjusted bacterial broths obtained from QST713, FZB24 and EU07 after 6, 24, 48 and 72 h of culture. Bacterial strains are shown at the top, left to right and by culture time on the left, top to bottom. (**B**) Percentage growth inhibition of the *Fg*-K1-4 by the Bacillus strains cultured at different times, filtered and adjusted. Broths adjusted to 100.0 µg/mL of proteins from filtered broths cultured at different times (24, 48 and 72 h). Data were from one independent experiment with four repeats and shown as the mean ± SE. The experiment was replicated three times with similar results. Bars clusters with different letters were significantly different according to Tukey’s Test (α < 0.05) following two-way ANOVA. n = 36.

**Figure 3 plants-11-01999-f003:**
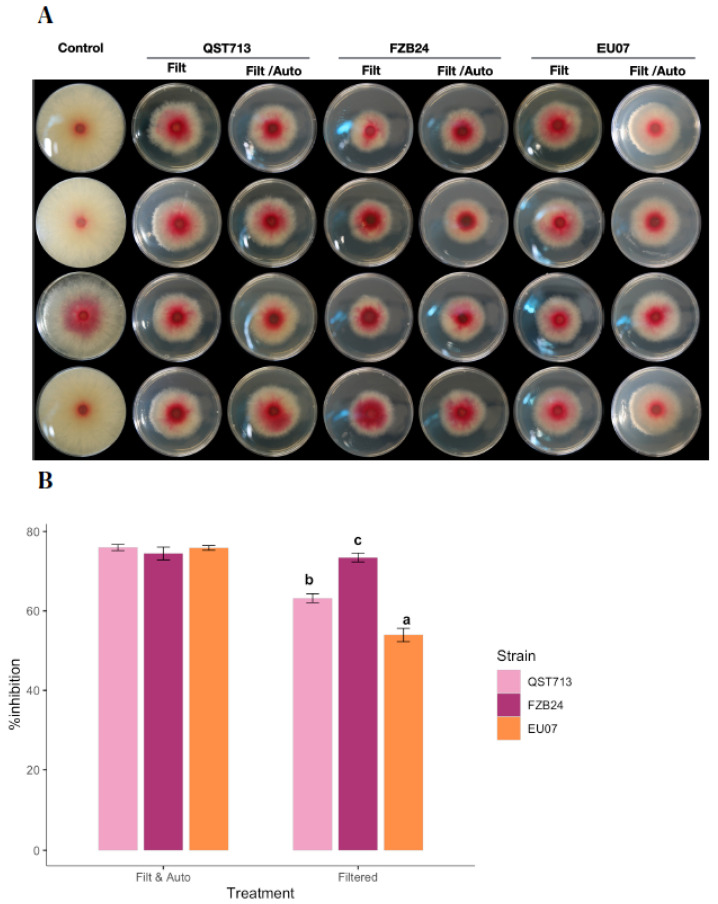
Inhibition assay with filtered unautoclaved or autoclaved bacterial broths. (**A**) Antagonism assays using filtered, autoclaved or unautoclaved broths. Bacterial broths were obtained from 24 h bacterial cultures, filtered, adjusted to 100 µg/mL and autoclaved. Bacterial broths are shown by bacterial treatment (top, left to right) and by type of treatment only filtered -Filt- and filtered/autoclaved -Filt/Auto- (top, left to right). (**B**) Percentage inhibition of the *Fg*-K1-4 by Bacillus broths that were filtered, adjusted and autoclaved. Antagonistic effect of the *Bacillus* strains (QST713, FZB24 and EU07) against *Fg-*K1-4 in % inhibition and broths adjusted (1.0 µg/mL). In general, the autoclaved treatments displayed a higher % inhibition with a little variability within *Bacillus* strains. Data were from one independent experiment of four repeats and shown as the mean ± SE. Experiments were replicated twice with similar results. Bars with different letters indicate significant difference according to Tukey’s Test (α < 0.05) following two-way ANOVA. n = 24.

**Figure 4 plants-11-01999-f004:**
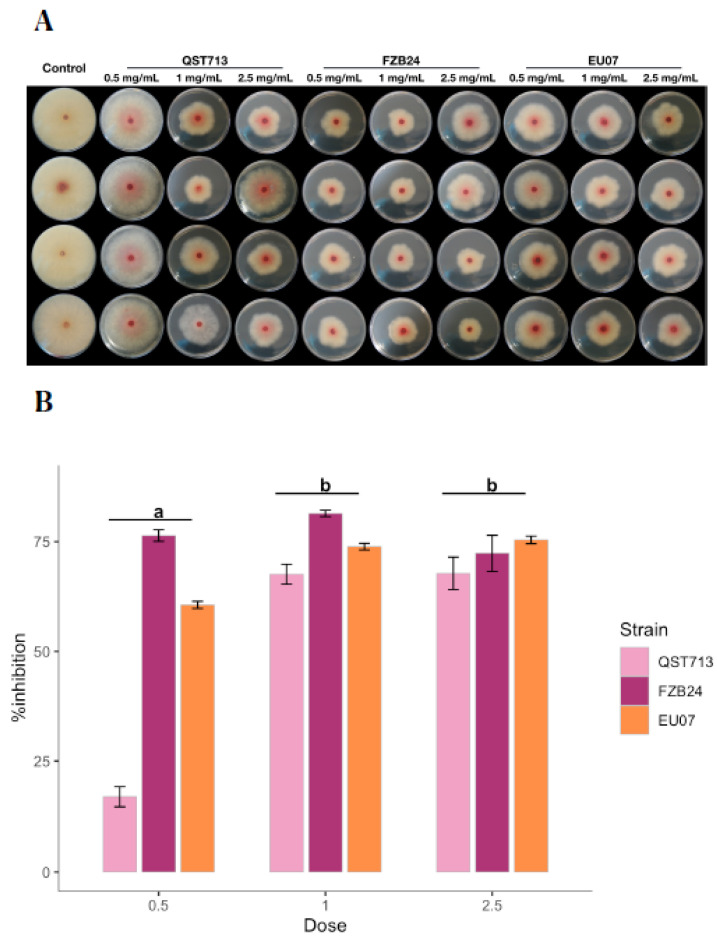
Effect of different bacterial broth concentrations on *Fg*-K1-4 growth. (**A**) Antagonism assays using filtered, adjusted and autoclaved broths. Bacterial broths were obtained from 24 h bacterial cultures, filtered, adjusted and autoclaved. Protein concentrations in bacterial broths were adjusted to 0.5, 1.0 and 2.5 mg/mL in the PDA/NA media. (**B**) Percentage inhibition of *Fg*-K1-4 by the *Bacillus* broths, which were filtered and adjusted to 0.5, 1.0 and 2.5 mg/mL of total protein in medium. Data were from one independent experiment of eight repeats and shown as the mean ± SE. Experiments were replicated twice, and the similar results were obtained. Bars clusters with different letters were significantly different according to Tukey’s Test (α < 0.05) following two-way ANOVA. n = 96.

**Figure 5 plants-11-01999-f005:**
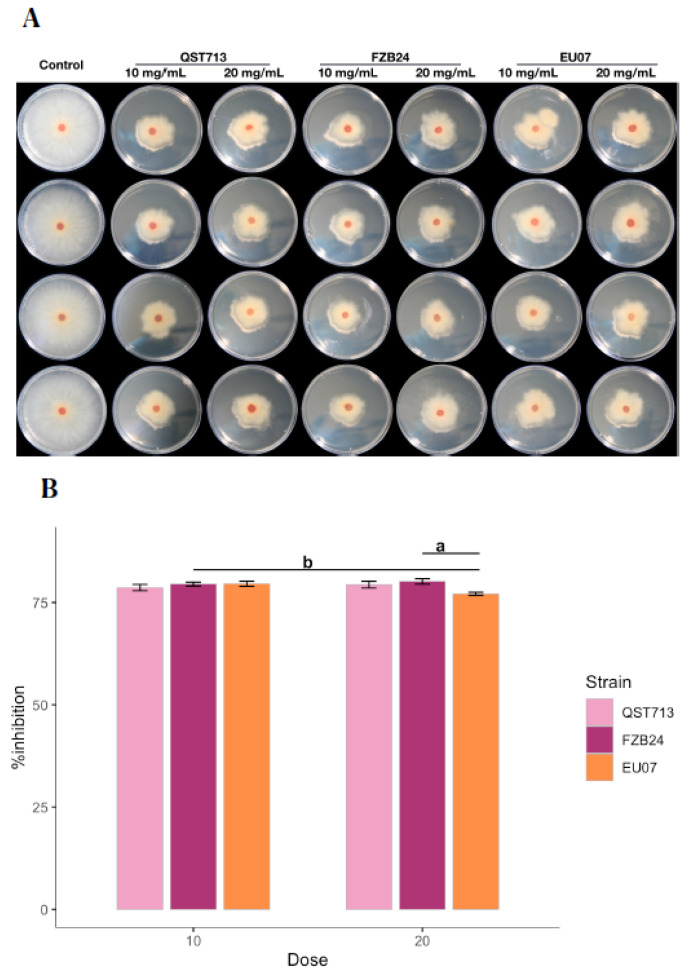
Effect of higher concentration of bacterial broths on *Fg*-K1-4 growth. (**A**) Antagonism assays on PDA/NA. Bacterial broths were obtained from 24 h bacterial cultures, filtered, adjusted and autoclaved. Bacterial broths are shown by bacterial treatment (top, left to right) and by dosage (top, left to right). High dosages of bacterial broths displayed low fungal growth compared with the control treatment (left column). (**B**) Percentage inhibition of the *Fg*-K1-4 by the broths of *Bacillus* strains (24 h), adjusted to 10.0 and 20.0 mg/mL of total protein in the media. Data were from one independent experiment of eight repeats and shown as the mean ± SE. Experiments were replicated twice, and the similar results were obtained. Bars with different letters were significantly different according to Tukey’s Test (α < 0.05) following two-way ANOVA. n = 48.

**Figure 6 plants-11-01999-f006:**
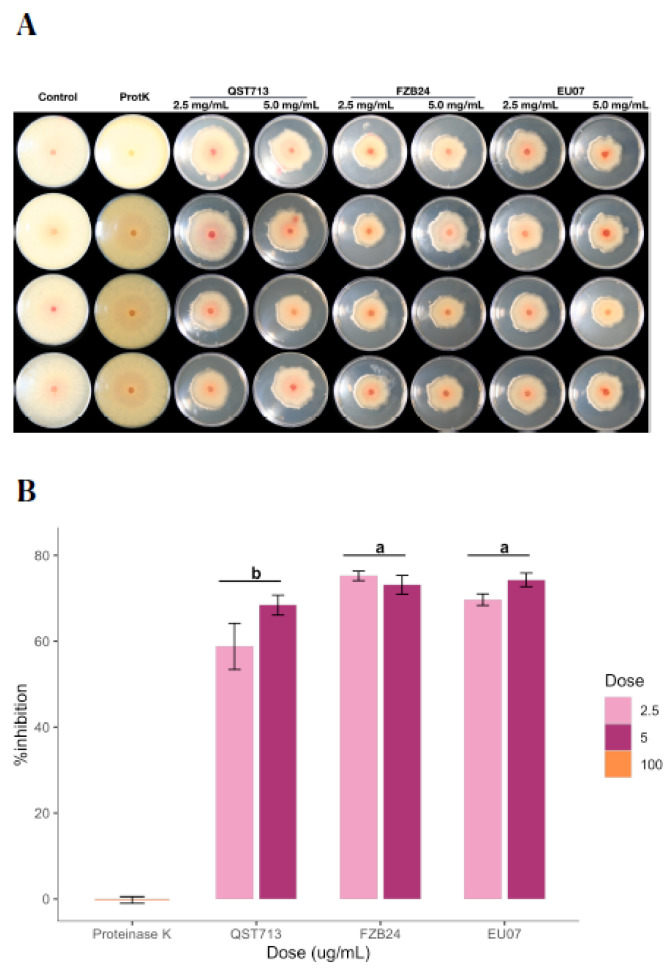
Antagonism assays using proteinase K treated bacterial broths. (**A**) Antagonistic effect of the bacterial broths filtered, autoclaved and proteinase K treated (100 µg/mL) of QST713, FZB24 and EU07 against *Fg-*K1-4 were performed on PDA/NA. Bacterial broths are shown by bacterial treatment (top, left to right) and by dosage (top, left to right). (B) Percentage inhibition of the *Fg*-K1-4 by the broths of *Bacillus* strains (24 h), adjusted to 2.5 and 5.0 mg/mL of total protein in the media. Data were from one independent experiment of four repeats and shown as the mean ± SE. Experiments were replicated two times and similar results were obtained. Bars clusters with different letters were significantly different according to Tukey’s Test (α < 0.05) following two-way ANOVA. n = 28.

**Figure 7 plants-11-01999-f007:**
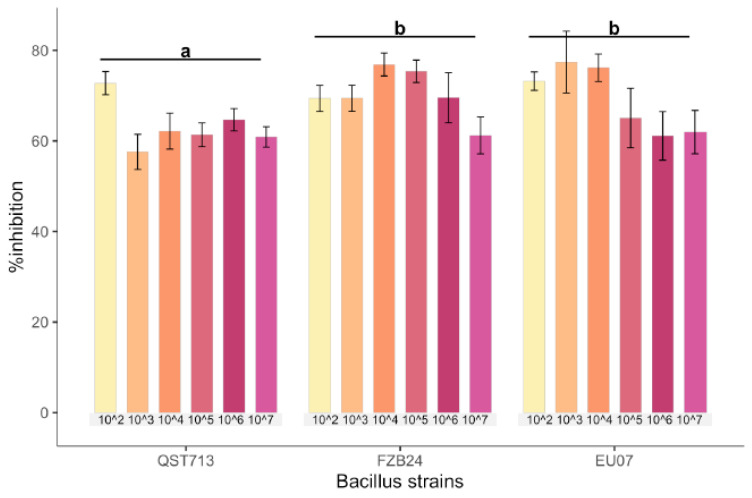
Growth of *Bacillus* strains in the presence of different DON concentrations. *Bacillus* strains (QST713, FZB24 and EU07) were grown in the presence of DON at different concentrations (0, 5, 15, 25, 50 and 70 ppm) in LB broth at 28 °C for 24 h at 200 rpm in 96-wells plates. Data were from one independent experiment with three replicates and shown as the mean ± SE. Experiments were repeated three times and similar results were obtained. Bar clusters with different letters were significantly different according to Tukey’s Test (α < 0.05) following two-way ANOVA. n = 72.

**Figure 8 plants-11-01999-f008:**
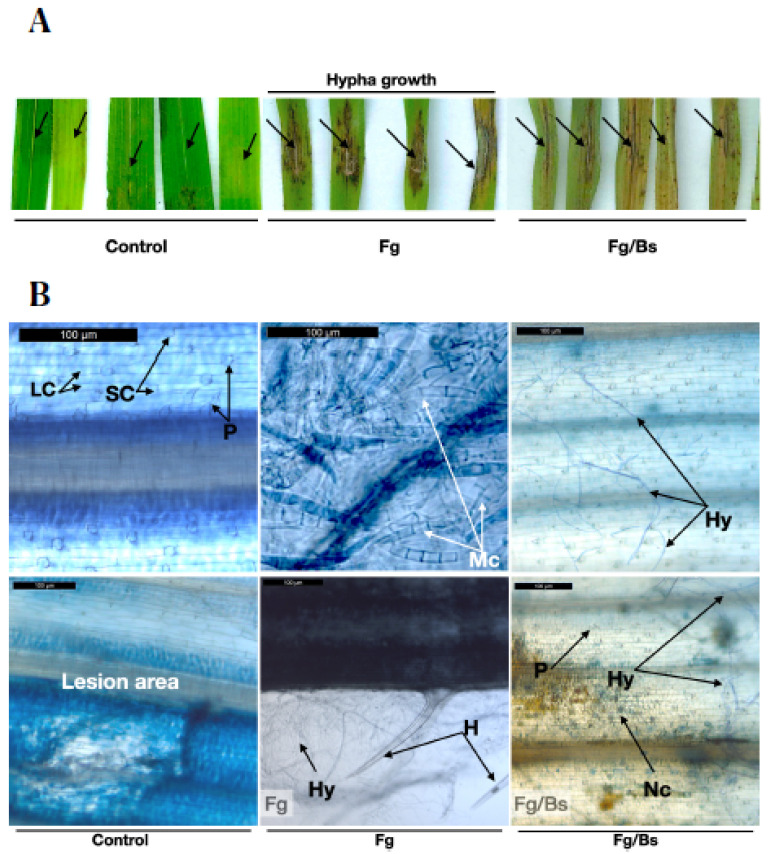
Detached leaves assay with Bd-21. (**A**) Bd-21 was used to assess the effect of the *Bacillus* on *Fg-*K1-4 on leaves 4d post inoculation (dpi). Detached leaves from ~2 months old plants were wounded using a glass rod, sown in water agar (WA) and inoculated with *Fg-*K1-4. Control (treated with water only) *Fg-*K1-4 infection shows dark necrotic areas around the inoculation point with conspicuous development of hypha. *Fg-*K1-4/EU07 treatment shows fewer necrotic areas around the laceration point with some mycelia growing from this point. (**B**) Trypan blue staining of the detached leaves of Bd-21 4 dpi. Adaxial view of leaves of Bd-21 non-infected (Control), infected with *Fg*-K1-4 (*Fg*) and infected with *Fg*-K1-4 and treated with *Bs* (Fg/Bs) Trypan blue. LC: Long cells, SC: Short Cells, P: Papilla, H: Hair, Hy: Hypha, Mc: Macroconidia, Nc: Necrotic damage. Scale bars = 100 µm.

**Figure 9 plants-11-01999-f009:**
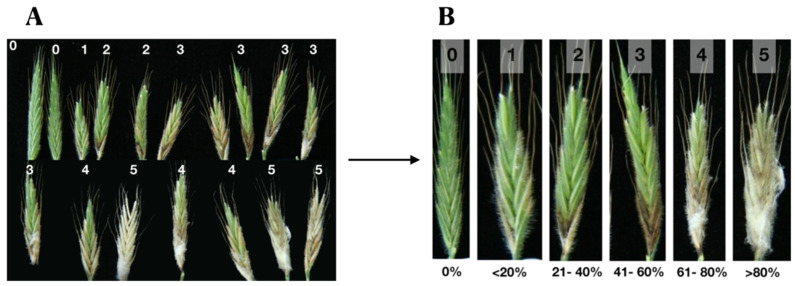
Construction of the severity scale for the infection head assay (IHA). Heads inoculated with *Fg*-K1-4. (**A**) Two weeks after inoculation, heads were collected, and a grade was assigned accordingly. (**B**) A total of six grades were made (0–5), which represents the % of progression of the disease and is called the Disease severity index (% DSI).

**Figure 10 plants-11-01999-f010:**
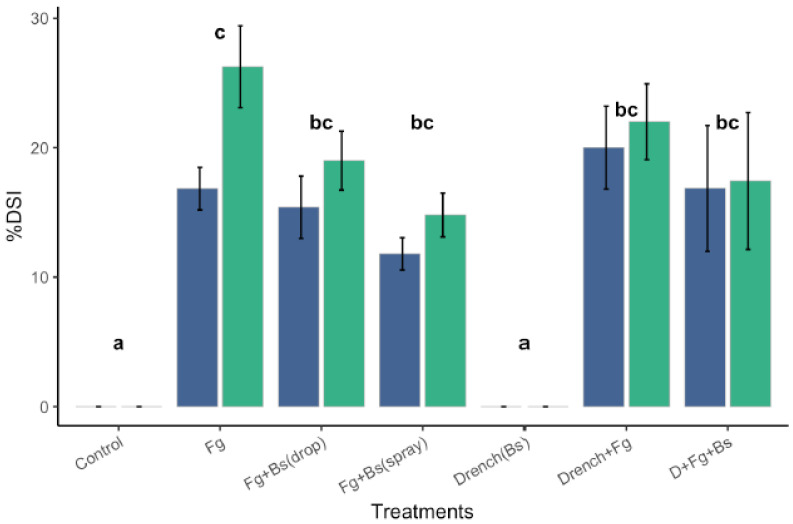
Disease severity index (DSI) of IHA with Bd-21 treated with *Fg*-K1-4 and EU07 at 7dpi and 14 dpi. Severity of infection of *Fg*-K1-4 on Bd-21 plants was determined for the Bd-21 plants treated and non-treated with *Bacillus* EU07 (*Bs*) and infected with or without *Fg-*K1-4. Spray Treatments using *Bs* to control *Fg-*K1-4 showed significant differences with the *Fg*-K1-4 treatments (drenched or not). Treatments applying drops or spraying *Bs* showed similar level of control over the *Fg-*K1-4. Data were from one independent experiment of at least seven repeats and shown as the mean ± SE. Experiments were replicated four times and similar results were obtained. Bars clusters with different letters were significantly different according to Tukey’s Test (α < 0.05) following two-way mixed ANOVA. n = 330. (*Fg*-K1-4 = *F. graminearum*, D = Drench, Bs = *Bacillus* strain EU07).

**Figure 11 plants-11-01999-f011:**
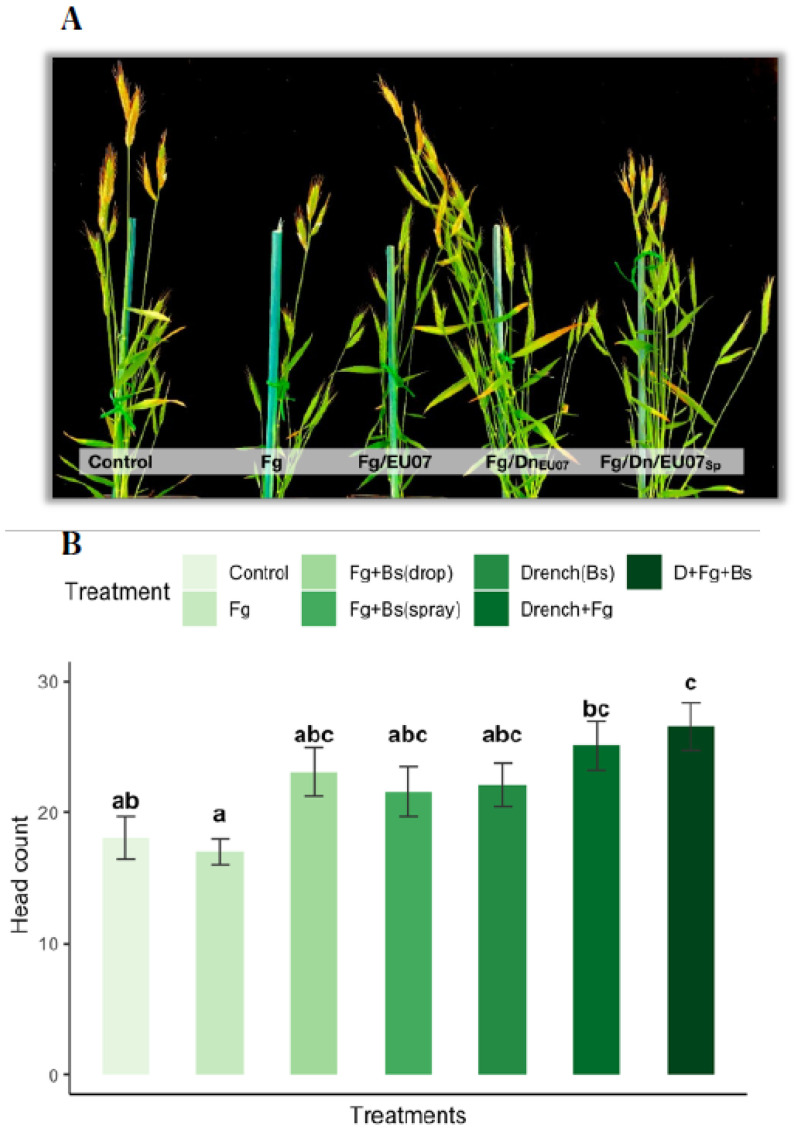
Bd-21 produced more spikes when treated with *Bacillus* EU07. (**A**) Bd-21 plants were inoculated with a 25 µL drop of macroconidia of *Fg* (10^6^/mL). The treatments were: Control (water), *Fg* alone, *Fg/EU07* inoculated and treated with *Bacillus* drop. *Fg*/Dn_EU07_: *Fg* inoculated plants and drenched with EU07. *Fg*/Dn/EU07_sp_: *Fg* inoculated plants, drenched with EU07 and treated with EU07 spraying the spikelets. Plants drenched with EU07 (*Fg*/Dn_EU07_ and *Fg*/Dn/EU07_sp_) produced a much higher number of spikelets than that by the control and the non-drenched plants. (**B**) Number of heads per plant of the Bd-21 treated with *Fg*-K1-4 and EU07. Data were from one independent experiment of at least seven replicates and shown as the mean ± SE. Experiments were repeated three times and similar results were obtained. Bars clusters with different letters were significantly different according to Tukey’s Test (α < 0.05) following one-way ANOVA. n = 66.

## Data Availability

The data that support the findings of this study are available from the corresponding author on reasonable request.

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
