# Peer review of "Comparison of Antifungal Activity of Bacillus Strains against Fusarium graminearum In Vitro and In Planta"

_plants, 2022, doi:10.3390/plants11151999_

Round 1
Reviewer 1 Report
Dear authors,
This studies are very interesting and indicates the strain EU07 is superior to other two commercial strains in suppressing F. graminearum and is a promising biocontrol agent against Fg-associated cereal diseases. Besides, in an environmental-friendly disease control in agriculture.
I highlighted in yellow in the text what I consider should be in the materials and methods section.

Reviewer 2 Report
Fusarium graminearum is a destructive fungal pathogen infecting multiple cereal crops worldwide. It has the ability to produce mycotoxins in contaminated cereal which are harmful to human health. For those reasons, the control of this fungal is essential and has been addressed for a long time ago. Biological controls are promising because they are environment friendly and have no residue effect on human health. In this study, the author assessed the antifungal ability of different Bacillus strains against the Fusarium graminearum in vitro and in planta. This study provides a potential resource and reference for further study and application in sustainable combat against the pathogen. I have some minor comments to improve the manuscript
- Line 64: the chemical controls are used many decades ago (not some years like the authors mentioned). Please find more accurate information and citation.
- Line 108: fix “IN this reserch” to “In this research”
- Line 217: remove the “medium” after 1.0 µg/mL
- Line 289: please define what the “a”, “b”, or “c” means when the authors classify levels of significance in statistic tests, as shown in the graphs
Author Response
- Line 64: the chemical controls are used many decades ago (not some years like the authors mentioned). Please find more accurate information and citation.
The word Chemical has been removed.
- Line 108: fix “IN this reserch” to “In this research”
Corrected
- Line 217: remove the “medium” after 1.0 µg/mL
Removed
- Line 289: please define what the “a”, “b”, or “c” means when the authors classify levels of significance in statistic tests, as shown in the graphs
The sentences below are inserted.
Letters were used to show the difference between the means. If two variables have different letters, they are significantly different.